# Screening of *Lactiplantibacillus plantarum* Strains from Sourdoughs for Biosuppression of *Pseudomonas syringae* pv. *syringae* and *Botrytis cinerea* in Table Grapes

**DOI:** 10.3390/microorganisms10112094

**Published:** 2022-10-22

**Authors:** Mariana Petkova, Velitchka Gotcheva, Milena Dimova, Elena Bartkiene, João Miguel Rocha, Angel Angelov

**Affiliations:** 1Department of Microbiology and Environmental Biotechnology, Agricultural University, 12 Mendeleev Blvd., 4000 Plovdiv, Bulgaria; 2Department of Biotechnology, University of Food Technology, 26 Maritza Blvd., 4002 Plovdiv, Bulgaria; 3Department of Phytopathology, Agricultural University, 12 Mendeleev Blvd., 4000 Plovdiv, Bulgaria; 4Department of Food Safety and Quality, Veterinary Academy, Lithuanian University of Health Sciences, Tilzes Str. 18, LT-47181 Kaunas, Lithuania; 5Institute of Animal Rearing Technologies, Faculty of Animal Sciences, Lithuanian University of Health Sciences, Tilzes Str. 18, LT-44307 Kaunas, Lithuania; 6Laboratory for Process Engineering, Environment, Biotechnology and Energy, Faculty of Engineering, University of Porto, 4050-345 Porto, Portugal; 7Associate Laboratory in Chemical Engineering, Faculty of Engineering, University of Porto, 4050-345 Porto, Portugal

**Keywords:** lactic acid bacteria (LAB), *Lactiplantibacillus plantarum*, plantaricins, antimicrobial activity, grapes *Vitis vinifera* L., *Pseudomonas syringae* pv. *syringae*, *Botrytis cinerea*

## Abstract

Grapes (*Vitis vinifera* L.) are an essential crop for fresh consumption and wine production. Vineyards are attacked by several economically important bacterial and fungal diseases that require regular pesticide treatment. Among them, *Pseudomonas syringae* pv. *syringae* (*Ps. syringae*) and *Botrytis cinerea* (*B. cinerea*) infections cause huge economic losses. The fresh fruit market has shifted to functional natural foodstuffs with clear health benefits and a reduced use of chemicals along the production chain. Lactic acid bacteria (LAB) have a biopreservative effect and are applied to ensure food safety in response to consumers’ demands. In the present study, the possibilities of using microorganisms with a potential antimicrobial effect against *Ps. syringae* and *B. cinerea* in the production of table grapes were investigated. LAB of the genus *Lactiplantibacillus* can be a natural antagonist of pathogenic bacteria and fungi by releasing lactic acid, acetic acid, ethanol, carbon dioxide and bacteriocins in the medium. The present study focuses on the characterization of nine *Lactiplantibacillus plantarum* (*Lp. plantarum*) strains isolated from spontaneously fermented sourdoughs. Species-specific PCR identified the isolated LAB for partial *recA* gene amplification with an amplicon size of 318 bp. RAPD-PCR analysis showed the intraspecific diversity of the individual strains. Thirteen plantaricin-like peptides (PlnA, PlnB, PlnC, PlnD, PlnEF, PlnG, PlnI, PlnJ, PlnK, PlnN, PlnNC8, PlnS, and PlnW) produced by isolated *Lp. plantarum* strains were detected by PCR with gene-specific primers. The key features for future industrial applications were their antimicrobial properties. The culture medium and cell-free supernatant (CFS) were used to establish in vitro antimicrobial activities of *Lp. plantarum* strains against *Ps. syringae* and *B. cinerea*, and inhibition of phytopathogen development was observed. The inhibitory effect of the CFS (cell-free supernatant) of all strains was assessed by infecting table grapes with these pathogens in in vivo experiments. *Lp. plantarum* Q4 showed the most effective suppression of the pathogens both in vitro and in vivo, which indicates its potential use as a biocontrol agent against berry rot and grey rot on grapes, caused by *Ps. syringae* and *B. cinerea*.

## 1. Introduction

Viticulture is one of the structural subsectors of Bulgarian agriculture with the potential to organize competitive and environmentally friendly production of high-quality grapes for fresh consumption and wines. Table grapes are highly appreciated by consumers, primarily because of their sensory attributes, but also because of their nutritional value and bioactive compounds [1,2,3]. The Italia variety occupies 1% of the total area for table grape cultivation in Bulgaria [4]. It is characterized by large fruits of an amber colour with a high content of sugars and organic acids. The muscat taste of the grapes appears relatively late, at the onset of maturity—at the consumer stage. The grapes are transportable but not suitable for storage. The attention of producers is focused on lengthening the shelf-life of table grapes to enable their exportation. Prolonged storage time would expand grape marketability and give added value, but the current methods to achieve it are often associated with a decrease in overall product quality. *Botrytis cinerea*, the causal agent of grey mould or botrytis bunch rot in grapes, is responsible for significant economic damage in vineyards worldwide [5]. Annual losses caused by *Botrytis* can range from 10 to 44% (pre- and postharvest), with the highest losses occurring in the postharvest stage by attacking buds, young shoots, leaves and grapes [6]. In conditions of high humidity, the bacterial pathogen *Pseudomonas syringae* pv. *syringae* causes bacterial inflorescence rot (BIR) with angular leaf lesions, longitudinal lesions in tissues and the rotting of inflorescences from before flowering until shortly after fruit set [7]. The necrotic bunch stem and leaf lesions caused by the bacterial pathogen were susceptible to the secondary development *of B. cinerea* infections [8]. Fruits are also infected through wounds caused by grape moth, wasps, hail, cracking after rain, etc. There is increasing interest in the use of biological control agents and plant resistance stimulants to suppress bacterial and fungal diseases in grapes [9]. The use of conventional synthetic antibiotics and fungicides is generating increasing concern among consumers due to the potential negative effects on human health [10], soil microbiota [11] and on microorganisms beneficial for food and beverage fermentations [12]. Consumers widely accept the development of bio-based applications to exert microbial control in the agri-food chain because of the growing demand for eco-friendly approaches and products free of synthetic chemicals [13,14].

Several compounds of biological origin have been assessed for their possible use as biological control agents against phytopathogens in table grapes [15,16,17,18]. Most of the microbial antagonists are isolated from traditional sources, such as sourdoughs [19,20,21,22], fermented dairy products [23], fruit and vegetable surfaces, roots and soil [24]. Those nonpathogenic microorganisms suppress the growth of plant pathogens through the competition for nutrients and the production of inhibitory metabolites and/or parasitism, thereby naturally limiting plant disease in the environment. Lactic acid bacteria (LAB) produce various antimicrobial substances during fermentation such as organic acids, hydrogen peroxide, carbon dioxide, and a wide range of low-molecular-weight compounds, peptides and proteins with antifungal and antibacterial potential [25,26,27,28,29,30,31]. Karpiński (2013) [32] reported that some LAB produce bacteriocins, which are ribosomally synthesized peptides, and exhibit a bacteriostatic or bactericidal activity against genetically closely related species. They differ from antibiotics in that they have a narrow spectrum of bioactivity and the organisms which synthesize bacteriocins have immunity against them [33,34]. Bacteriocins are of great importance for food preservation. In 2006, nisin and pediocin PA1 (MicrogardTM, ALTA 2431, Quest) were the only bacteriocins classified as generally recognized as safe (GRAS) and used as food preservatives in several countries [35]. Vescovo et al. (1996) [36] and Torriani et al. (1997) [37] reported for the first time the utilization of *Lacticaseibacillus casei* and its culture filtrate to control pathogenic microorganisms in ready-to-use vegetables. An antagonistic effect of *Pseudomonas graminis* CPA-7 against two foodborne pathogens (*Salmonella* spp. and *Listeria monocytogenes*) was recorded in fresh-cut apples and melons [38,39].

Several strains of bacteriocin-producing *Lp. plantarum* have been isolated in the last two decades from different ecological niches including meat, fish, fruits, vegetables, milk and cereals [40,41]. However, many of bacteriocins produced by different *Lp. plantarum* strains have not been fully characterized [42]. Genes that encode Class II bacteriocins are organized in clusters [33]. Plantaricins belong to Class IIb of bacteriocins, as they are small (30–100 amino acids), thermostable and commonly not post-translationally modified. In addition, they are generally subdivided into four subclasses: a, b, c and d. Moreover, it has been shown that several LAB produced multiple bacteriocins [33,34]. A common characteristic is a conserved YGNGV motif and a disulphide bond linkage in the N-terminal region that is essential for the strong inhibitory effect on food spoilage and pathogenic bacteria [43]. Class IIb consists of complexes requiring two peptides for their activity (lactococcin G and F, lactacin F, and plantaricin EF and JK). They consist of a structure gene encoding for the prepeptide, a dedicated immunity gene [34], an ABC transporter gene transport across the membrane [44] and the gene encoding to the accessory protein for export of the bacteriocins [45]. Bacteriocins from *Lp. plantarum* strains isolated from fermented foods have been characterized for their use as natural food preservatives [46,47,48,49]. *Lactiplantibacillus plantarum* C11 secretes a small cationic peptide, plantaricin A, which serves as an induction signal for bacteriocin production as well as for transcription of plnABCD [34]. The *plnABCD* operon encodes the plantaricin A precursor (*plnA*) itself and the determinants (*plnBCD*) for a signal-transducing pathway [34]. Four new plantaricin A-induced operons were identified and all were highly activated in concert with *plnABCD* upon bacteriocin induction. Two of these operons, *plnEFI and plnJKLR*, encompass a gene pair (*plnEF* and *plnJK*, respectively) encoding two small cationic bacteriocin-like peptides with double-glycine-type leaders. On the third operon, termed *plnMNOP*, a similar bacteriocin-like *plnN* and a putative immunity *plnM* were identified as well. These findings suggest that two bacteriocins of a two-peptide type, *plnEF* and *plnJK*, and a one-peptide-type bacteriocin, *plnN*, could be responsible for the observed bacteriocin activity. The last operon, *plnGHSTUV*, contains two open reading frames of *plnGH* apparently encoding an ABC transporter and its accessory protein, which is known to be involved in the processing and export of peptides with precursor double-glycine-type leaders. Furthermore, bacteriocins active against *Escherichia coli*, *Salmonella* and *Pseudomonas aeruginosa* have been produced in planta with the aim for them to be used as food additives [50]. Moreover, it has been recently demonstrated that bacteriocins active against plant pathogenic bacteria can be expressed in plants to provide robust resistance against plant disease [51]. In general, the antibacterial and antifungal activity of LAB appears to be correlated to metabolic products that can also act in synergy. The chemical nature and the produced amount of these compounds are species- and strain-dependent and include organic acids [52,53,54].

In the present study, LAB strains have been selected for screening as potential biological agents for the control of *Ps. syringae* and *B. cinerea.* The antifungal properties of bactericidal *Lp. plantarum* strains isolated from Bulgarian sourdough were examined. The LAB strains were subjected to molecular screening for the presence of plantaricin genes, and the capacity of bacteriocins produced by the tested isolates to inhibit *Ps. syringae* pv. *syringae* and *B. cinerea* was evaluated in vitro.

## 2. Material and Methods

### 2.1. LAB Strains and Plant Pathogens

LAB strains used in this study were chosen from the collection of the Department of Microbiology and Environmental Biotechnology, Agricultural University of Plovdiv. These LAB strains were previously isolated from spontaneously fermented sourdoughs [19]. *Ps. syringae* pv. *syringae* is pathogenic to a large number of horticultural plant hosts worldwide, including viticulture [7,55]. The pathogenic bacteria *Ps. syringae* pv. *syringae* and fungus *B. cinerea* were isolated from damaged grape leaves and grapes at the Department of Phytopathology, Agricultural University of Plovdiv. The isolation was carried out after disinfection of the material with 70% (v/v) ethanol, and it was crushed using a mortar and pestle in sterile distilled water. *Ps. syringae* pv. *syringae* was cultivated on King’s medium B [56] at 26 °C for 24 h. To confirm that *P. syringae* pv. *syringae* was responsible for the disease development on grapevines, Koch’s postulates were tested by inoculating leaves of *V. vinifera* Italia variety plants with the isolated strain of *P. syringae* pv. *syringae* and maintaining the leaves under humid conditions.

*B. cinerea* was cultured on yeast extract medium (5 g/L yeast extract, 10 g/L glucose and 20 g/L agar) at 25 °C for 7 days. Then, the conidia were collected by washing with cold, sterile, distilled water and used to prepare an inoculum at a concentration of 1 × 10^4^ spores/mL. The concentration of spores in the suspension was determined by counting the spores in a Thoma counting chamber and calculating their concentration in the initial suspension. Grapes were infected with the spore suspension and incubated in individual moist chambers.

### 2.2. Plant Material

Well-ripened and uniform in size grapes (*Vitis vinifera* L.) of the Italia variety were provided for the study by the Department of Viticulture at the Agricultural University of Plovdiv.

### 2.3. Molecular Analyses

#### 2.3.1. PCR Analysis with *recA* Gene

The 24 h biomass of the cultures in MRS broth (Oxoid^®^, Hampshire, UK) was washed twice and resuspended in 100 μL of DNase-free deionized water (Merck KGaA, Darmstadt, Germany). DNA isolation from LAB cultures was performed with the E.Z.N.A.^®^ Bacterial DNA Kit (Omega Bio-Tek, Inc., Norcross GA, USA), according to the manufacturer’s instructions. The Gram-positive bacteria lysis buffer was prepared by mixing 20 mM Tris-HCl (Merck KGaA, Darmstadt, Germany), pH 8.0; 2 mM EDTA (Merck KGaA, Darmstadt, Germany); 1.2% (*v*/*v*) Triton X-100 (Merck KGaA, Darmstadt, Germany); and 20 mg/mL lysozyme (Merck KGaA, Darmstadt, Germany). Following the manufacturer’s protocol, 10 μL of proteinase K and 5 μL of ribonuclease A were added (Thermo Fisher Scientific, Waltham, MA, USA). The quality of DNA was determined by an agarose gel electrophoresis system (VWR GmbH, Darmstadt, Germany) by using 0.8% (*w*/*v*) agarose gel. DNA concentration was determined by measuring the absorbance of the samples at 260 nm in a UV–Vis 1800 spectrophotometer (Shimadzu, Tokyo, Japan). The purity of the DNA in the extracts was determined by calculating the ratio between the absorbance of the samples at 260 nm and 280 nm and between the absorptions at 260 and 230 nm.

Isolated DNA from the strains selected after the microbiological analysis was used in polymerase chain reaction (PCR) amplification with *Lactobacillus*-specific genes based on the *recA* gene, planF (5′-CCG TTT ATG CGG AAC ACC TA-3′) and pREV (5′-TCG GGA TTA CCA AAC ATC AC-3′) from the method of Torriani et al. (2001) [57]. The PCR reaction mixture consisted of two microliters of 10 × PCR buffer, 2 µL of 2.5 mM dNTPs, 2 µL of 2.5 mM MgCl_2_ solution, 1 μL of 10 μM primers and 0.25 μL of 5 U/μL Red-Taq DNA Polymerase (Canvax, Cordoba, Spain). Amplification was performed on a PCR 2720 Thermal Cycler (Thermo Fisher Scientific Waltham, MA, USA) using the following program: initial denaturation at 95 °C for 10 min, followed by 35 cycles of denaturation at 94 °C for 30 s, multiplication at 50 °C for 1 min, extension at 72 °C for 2 min and a final extension at 72 °C for 7 min. PCR products were visualized in 1.5% (*w*/*v*) agarose gel stained with Safe View (NBS Biologicals, Huntingdon, UK) at 100 V for 50 min using a VWR Mini Electrophoresis System (VWR, Vienna, Austria) and a MiniBis Pro (DNR Bio-Imaging Systems, Neve Yamin, Israel) for gel visualization. The Gene Ruler 1 kb Plus (Bioneer, Seoul, S. Korea) was used as a molecular marker. A reference strain *Lactiplantibacillus plantarum* ATCC 8014 was purchased from Ridacom Ltd. (Sofia, Bulgaria).

#### 2.3.2. Genotyping of LAB Isolated from Sourdoughs by RAPD-PCR

Along with species determination, an important stage in studying the microbiota of different sourdough samples was the characterization of their intraspecific diversity. For this purpose, a genotyping method with a proven high discriminatory ability was used. RAPD includes random amplification from the target bacterial genome using single primers under less stringent conditions. Chromosomal DNA from nine different isolated *Lp. plantarum* strains was used as a PCR template with primer P1 (5′-GCGGCGTCGCTAATACATGC-3′), P4 (5′-CCGCAGCGTT-3′) and M13 (5′-GAGGGTGGCGGTTCT3′) [58,59,60]. The assay based on the detection of small inverse nucleotide sequences throughout the genomic DNA of the isolated strains of *Lp. plantarum* was performed using the method of Ehrmann and Vogel (2005) [58]. PCR products were visualized in 2% (*w*/*v*) agarose gel stained with Safe View (NBS Biologicals, Huntingdon, UK) at 100 V for 50 min using a VWR Mini Electrophoresis System (VWR, Vienna, Austria) and MiniBis Pro (DNR Bio-Imaging Systems, Neve Yamin, Israel) for gel visualization. The Gene Ruler 1 kb Plus (Bioneer, Seoul, S. Korea) was used as a molecular marker. The results of the RAPD-PCR profiles were combined and analysed to obtain a unique dendrogram. Comparison of the obtained profiles was performed by calculating an index of genetic similarity using the coefficient of similarity calculated from Simple Matching by the VWR system software (VWR, Leuven, Belgium). Cluster analysis was performed using the UPGMA method, from which a dendrogram showing the linkages between different strains of *Lp. plantarum* was obtained.

#### 2.3.3. Screening of *Lp. plantarum* Strains for the Production of Known Plantaricin Genes

Polymerase chain reaction (PCR) analysis for the screening of 13 genes from *pln*A to *pln*W encoding the production of plantaricins was performed in 25 µL for each reaction, with each volume containing 50–100 ng DNA of the *Lp. plantarum* strains. The total volume of the PCR reaction was 20 µL. The mixture consisted of 2 µL of 10 × PCR buffer, 2 µL of 2.5 mM dNTPs, 2 µL of 2.5 mM MgCl_2_ solution, 1 μL of 10 μM primers and 0.25 μL of 5 U/μL Red-Taq DNA Polymerase (Canvax, Cordoba, Spain). PCR analysis for genes involved in plantaricin production *plnA, plnB, plnK*, *plnNC8*, and *plnW* was performed under the following conditions according to the protocol of Omar et al. (2008) [61]: 35 cycles of initial denaturation at 95 °C for 5 min, denaturation at 95 °C for 30 s, hybridization at 53 °C for 1 min, extension at 72 °C for 30 s and elongation at 72 °C for 3 min. For genes *pln*C, *pln*G and *plnJ*, the initial denaturation was performed at 95 °C for 5 min, denaturation at 95 °C for 30 s, hybridization at 51 °C for 1 min, extension at 72 °C for 30 s, followed by final elongation at 72 °C for 3 min. For genes *plnEF, plnN*, *plnS,* and *plnC*, the conditions were: initial denaturation at 95 °C for 5 min, denaturation at 95 °C for 30 s, hybridization at 52 °C for 1 min, extension at 72 °C for 30 s and prolonged elongation at 72 °C for 3 min [62,63]. For *plnI*, the conditions were: initial denaturation at 95 °C for 5 min, denaturation at 95 °C for 30 s, annulation at 55° C for 1 min, extension at 72 °C for 30 s and elongation at 72 °C for 3 min. The amplicons were separated into 2% (*w*/*v*) vegetable gel with Safe View (NBS Biologicals, Huntingdon, UK) at 100 V for 50 min using the VWR Midi Electrophoresis Imaging System (VWR GmbH, Darmstadt, Germany). The Gene Ruler 1 kb Plus (Bioneer, Seoul, South Korea) was used as the marker.

### 2.4. In Vitro Antimicrobial Activity of LAB Strains

#### 2.4.1. Antibacterial Activity Screening of Cell-Free Supernatants (CFSs) from LAB against *Ps. syringae* pv. *syringae*

The antibacterial activity of the nine bacteriocinogenic *Lp. plantarum* strains was tested by the agar-well diffusion method, according to the previous research of Petkova et al. (2021) [20]. Cell-free supernatants (CFSs) were obtained by centrifugation for 24 h of MRS-broth cultures of the tested LAB strains. The CFSs were consecutively subjected to the following treatments: adjustment of pH to 6.5, boiling for 20 min, addition of catalase (5 mg/mL) (Merck, Darmstadt, Germany) and treatment with trypsin (1 mg/mL) (Merck, Darmstadt, Germany). The pathogenic bacterium *Ps. syringae* pv. *syringae* was isolated from damaged grapes in the Department of Phytopathology, Agricultural University of Plovdiv, according to Hall et al. (2016) [64]. *Ps. syringae* was cultivated on PDA (HiMedia, Thane West, Maharashtra, India) at 30 °C for 24 h. After inoculation of the 9 mm wells with 100 µL of the respective CFSs, the plates were incubated at 30 °C for 48 h and the areas of growth inhibition were measured.

#### 2.4.2. Antifungal Activity Screening of Cell-Free Supernatants (CFSs) from Tested LAB against *B. cinerea*

The LAB strains showing strong antifungal activity were inoculated in MRS broth and incubated at 30 °C for 48 h. CFSs were obtained after centrifugation at 13,000× *g* for 10 min and sterilized through a sterile 0.22 µm-pore-size filter unit (Sigma, Steinheim, Germany). To find out if the antimicrobial compounds were of a proteinaceous nature, cell-free supernatants of isolates with antimicrobial activity were treated with 1 M NaOH for the adjustment of the pH to 6.5 and subjected to boiling for 20 min, or to a treatment with catalase (5 mg/mL) (Merck KGaA, Darmstadt, Germany), trypsin (1 mg/mL) (Merck KGaA, Darmstadt, Germany) and proteinase K (Merck KGaA, Darmstadt, Germany). Afterwards, the antifungal activity was determined by an agar dilution assay based on the method described by Russo et al. (2017) [13]. After treatment with those enzymes, each CFS was added at a volume of 700 μL in each PDA (HiMedia, Thane West, Maharashtra, India) and incubated at 4 °C for 3 h. Further, the plates were incubated with 8 mm discs of 5-day-old *B. cinerea* culture. The MRS medium (700 μL) was used as a control. The plates were incubated at 28 °C for 5–7 days and the areas of growth of the fungal pathogen were measured according to the method of Ogunremi et al. (2022) [65]. The antibacterial activity of the CFSs was analysed in three independent experiments.

### 2.5. In Vivo Antimicrobial Activity of Lp. plantarum Strains

The inhibitory effect of *Lp. plantarum* on the phytopathogens was evaluated in in vivo experiments by using healthy and well-ripened grapes of the Italia variety, grown in the experimental field of the Agricultural University of Plovdiv, which is located in Brestnik village, Bulgaria (42°44′29.00″ N 22°53′53.02″ E). Each assay was performed with 80 grapes identical in shape, size and colour, which were divided into four groups of ten in different boxes as two replicas. Their surfaces were sterilized to remove the epiphytic microbiota following the standard protocol by Tryfinopoulou et al. (2015) [66]. The plant samples were washed with running tap water to remove the adhering dust and debris. Surface sterilization was performed by immersion in 96% ethanol for 1 min to remove the adhering microorganisms. The fruits were finally rinsed with deionized, sterile, distilled water to remove the chemical agents.

#### 2.5.1. In Vivo Antibacterial Activity of *Lp. plantarum* Q4 against *Ps. syringae* pv. *syringae* in Grapes

The ability of *Lp. plantarum* Q4 to inhibit *Ps. syringae* was investigated using artificially injured grapes, according to the method of Chen at al. (2022) [67]. The grapes were selected with similar size, shape, colour, weight and absence of mechanical damage and fungal infection. Each grape was injured with a sterile perforator to make uniform wounds 2 mm deep and 2 mm wide on its equatorial region. Wounded fruits treated with *Lp. plantarum* Q4 cell-free supernatant were stored at 4 °C for 10 h before inoculation of the bacterial pathogen. Then, 20 µL suspensions of pathogen spores (10^6^ spores/mL) were inoculated into the wounds of each grape. The percentage of bacterial infection of fruits was reported on the 3rd and 5th day after infection.

#### 2.5.2. In Vivo Antimicrobial Activity of *Lp. plantarum* Q4 against *B. cinerea* in Grapes

The ability of *Lp. plantarum* Q4 to inhibit *Botrytis cinerea* was investigated using artificially injured grapes [68]. Each grape was injured with a sterile perforator to make uniform wounds 2 mm deep and 2 mm wide on its equatorial region. Wounded fruits were treated with *Lp. plantarum* Q4 cell-free supernatant and then stored at 4 °C for 10 h before inoculation with the fungal pathogen. Then, 20 µL suspensions of pathogen spores (10^4^ spores/mL) were inoculated into the control grapes and the wounds of each test grape. The concentration of spores in the initial suspension was determined by counting the spores in a Thoma counting chamber and calculating their concentration in the initial suspension. The percentage of fungal infection of fruits was reported on the 3rd and 5th day after infection.

### 2.6. Statistical Analysis

The results from the in vitro and in vivo testing of the antimicrobial activity of the LAB strains were expressed as the mean (*n* = 3) ± standard deviation (SD). Statistica 7.0 software (Version 7.0; accessed on 22 January 2022) was used for the statistical evaluation. A *p* ≤ 0.05 was considered statistically significant. Statistics were performed with SPSS for Windows XP V15.0 (SPSS, Inc., Chicago, IL, USA, 2007).

## 3. Results and Discussion

### 3.1. Plans Material and Plant Pathogenicity Tests

Italia is a typical table grape variety, with fruits ripening in the second half of September. The variety was developed by hybridising Bican x Muscat Hamburg varieties in Pirano (Italy) in 1911. It is found in almost all wine-growing countries. In Bulgaria, it occupies 1% of the area of table grape vineyards. The duration of the period from budding to consumer maturity is 160 days. The variety is sensitive to drought and relatively resistant to low temperatures. It was chosen for our study because of its susceptibility to pests and fungal diseases, especially powdery mildew and grey rot.

*Ps. syringae* and *B. cinerea* were used in the plant pathogenicity tests for the infection of the vine leaves and grapes. Figure 1A displays the state of a control vine leaf. After infection with *Ps. syringae*, the pathogen caused necrotic lesions surrounded by halo blight spots around them after 5–7 days of incubation (Figure 1B). *B. cinerea*-infected berries first appeared soft and watery and then became brown. The pathogen caused a grey-brown sporulating growth of the fungus on the surface of infected fruits during the 5–7 days of incubation (Figure 1D). Re-isolations were made from laboratory-infected plants. Re-isolations were established from leaves and grapes and the developing disease symptoms were observed (Figure 1B,C). The results of isolation and re-isolation of the two pathogens coincided, proving their pathogenicity.

### 3.2. Molecular Analysis of Lp. plantarum Strains

#### 3.2.1. Microbiological Analysis

Bacterial isolates were identified using species-specific primers for *Lactiplantibacillus plantarum*, according to Torriani et al. (2001) [57]. The PCR amplification yielded a single fragment of 328 bp, corresponding to the profile of lactic acid bacteria of the *Lp. plantarum* group (Figure 2). The PCR product was found in both the DNA extracted from the positive control (*Lp. plantarum* ATCC 14917T, Ridacom, Sofia, Bulgaria) and the DNA extracted from the nine strains tested in the current experiment. Previous studies on the microbiology of sourdough showed that *Lp. plantarum* was dominant in the traditional sourdoughs from the Balkan region, Italy and fermented African foods [69,70,71]. The main aim of the present study was focused on bacteriocin production and antimicrobial activities of strains isolated from Bulgarian sourdoughs from the Rhodope mountain region in South Bulgaria.

#### 3.2.2. RAPD-PCR Analysis

Various techniques for molecular typing of microorganisms at species and strain level have been developed and successfully applied over the last decades. These methods rely on revealing polymorphisms in the DNA profiles of species and/or strains of a species and have high discriminatory power, reproducibility, ease of interpretation and standardization. RAPD-PCR analysis is a fast, sensitive and relatively inexpensive method providing excellent results [60]. To study the genotype diversity among the *Lp. plantarum* strains, a RAPD-PCR analysis was performed. After the successful usage of three primers, M13, P1 and P4, a total of 146 specific RAPD profiles were compared with the initial treatment of isolated *Lp. plantarum* strains and grouped based on the degree of similarity (Figure 3a–c).

Primer P1 yielded two different patterns of bands (Figure 3a). The pattern with both 600 and 700 bp bands was the most common (present in 66,6%) among the strains (C6, I6, K1, L1, Q3 and Q4) isolated from sourdough samples. DNA from strains D16, S1 and S4 produced only a 700 bp fragment. Primer P4 yielded a number of band patterns. Four different patterns with primer P4 were observed (Figure 3b). According to this PCR result, RAPD patterns were assembled in three groups. K1, L1 and Q3 had a similar P4 pattern. The second group included D16, C6, I6 and S6. Q4 and S1 produced bands with primer P4 which differ from the other LAB. The RAPD profile with primer M13 was identical for strains I6, K1, L1 and Q3 (Figure 3c). Strains Q4 and S1 clustered together. Conversely, the other three RAPD patterns of D16, C6 and S4 were clearly unrelated to the other *Lp. plantarum* strains.

The obtained phylogenetic data were subjected to UPGMA analysis, based on which a dendrogram illustrating the genetic similarity among the strains was constructed (Figure 4). The dendrogram shows the presence of intraspecies diversity among the nine *Lp. plantarum* species.

#### 3.2.3. Detection of Plantaricin Production Genes

In an attempt to determine whether the selected strains carry genes for the production of known bacteriocins, primers specific for individual plantaricin genes were used in the PCR analysis (Table 1). Nine isolates identified as *Lp. plantarum* were tested for genes of the plantaricin cluster, previously described by Diep (1996, 2003) [34,72]. Plantaricin synthesis genes were identified in all isolates of the present study (Table 1; See also Appendix A).

In the current study, PCR analysis showed the presence of plantaricin A (*plnA*) in *Lp. plantarum* strains C6, K1, L1, Q3, Q4, S1 and S6. *plnA* induces the transcription of genes organized by five operons: *plnABCD*, *plnEFI*, *plnJKLR*, *plnMNOP,* and *plnGHSTUV* [34,61,72,73,74]. *plnA, plnE, plnF, plnJ, plnK, and plnN* encode typical bacteriocyte peptide precursors, whereas *plnI* has been suggested to encode immunity-related proteins [34]. Other genes commonly found along with bacteriocin production were *plnB* and *plnC*, which encode proteins involved in the signal transduction of pheromones [75]. In our study, *plnB* was amplified from DNA from all examined strains. *plnC* was not found when D16, L1 and Q3 were used as a matrix in the PCR reaction. On the other hand, *plnD* was detected in all isolated *Lp. plantarum* strains. Variations in this operon have been reported also for gene clusters described in experiments with *Lp. plantarum* NC8 from olives [63] and *Lp. plantarum* J23 from grape must origin [76]. Comparative genomic analysis of *Lp. plantarum* strains also suggested high variability in plantaricin production and immunity [74]. Therefore, similar variants may also occur in the strains obtained in the present study.

Plantaricin EF and plantaricin W, encoded by *plnEF* and *plnW* loci, are classified as class I and class II bacteriocins, respectively [72]. The presence of two loci-encoding bacteriocins of different classes contributes significantly to the broad inhibitory spectrum of *Lp. plantarum* strains [74]. PlnW is a novel two-peptide bacteriocin from *Lp. plantarum* that inhibits a large number of Gram-positive bacteria [77]. *plnEF* is present in the genome of all nine LAB strains studied. However, *plnW* was found only in the isolate D16. When tested against the Gram-negative model organism *Escherichia coli* K-12 the *plnJ*- and *plnK*-producing lactic acid bacteria showed high efficacy under certain conditions [62]. In the present study, *plnJ* screening by PCR was performed with specific primers, according to Omar et al. (2008) [61]. The result was negative only for strain I6. *plnK* was amplified in the genome of the C6, L1, Q3, S1 and S4 strains. A previous study showed that *plnD*, *plnEF*, *plnI*, *plnG,* and *plnK* were the most common genes in lactic acid bacteria [61]. Although *Lp. plantarum* is widespread in fermented foods, genes encoding *plnNC8* and *plnW* were not detected in any strain isolated from reported samples [63,75,77]. The abundance of *plnEF, plnJ* and *plnI* is the main similarity in published studies, which was confirmed by our results. Holo et al. (2001) [77] and Maldonado et al. (2002) [63] did not detect plantaricin *plnNC8* and *plnW*. However, in the present study, *plnNC8* was detected in all strains but *Lp. plantarum* S1. The *plnS* gene was found in approximately 20% of the isolated lactic acid bacteria from dairy products in two different studies [78]. In the molecular study, *plnS* was found in *Lp. plantarum* D16, I6, K1, L1, Q3, Q4 and S4 strains. The production of bacteriocins is often linked to mobile genetic elements that can facilitate the transfer of genes between different species and strains sharing the same niche [72,73]. The presence of genes encoding the secretion of plantaricins also suggests the ability to inhibit the development of pathogenic microorganisms.

### 3.3. Antimicrobial Activity of LAB Strains

#### 3.3.1. Determination of the Spectrum of Antibacterial Activity of *Lp. plantarum* Strains against *Ps. syringae*

The antibacterial activity of nine *Lp. plantarum* strains isolated from sourdough was determined against the indicator strain of *Ps. syringae* by the well diffusion method. When evaluating the probiotic properties of each strain, the activity of CFSs should not be ignored since their biological activity in vivo will always occur in the presence of the LAB natural metabolites. The results in Table 2 show that CFSs of all tested *Lp. plantarum* strains suppressed the growth of pathogenic bacteria.

To demonstrate the protein nature of the inhibitory agents obtained after neutralization and elimination of hydrogen peroxide, cell-free supernatants were treated with proteolytic enzymes—trypsin, proteinase K, and catalase. The highest inhibitory effect was observed with the use of the cell-free cultural supernatant (CFS) of *Lp. plantarum* Q4, followed by *Lp. plantarum* C6, I6, D16 and Q3 (Table 2). The lowest inhibitory effect of the bacterial pathogen was detected with the CFS of *Lp. plantarum* S4 and K1 strains. The treatment with NaOH totally eliminated the antibacterial activity of strains D16, L1 and S4. These results prove that the inhibitory effect of both strains was due to the release of lactic acid and the subsequent acidification of the medium. Adjusting the pH to 6.5 led to a decrease in the antimicrobial activity of the strains D16, C6, K1, Q3 and S1. This is in contrast to the results of Haghshenas et al. (2014) [78], who observed that increasing the initial pH of the culture medium could enhance the bacteriocin production of *Lactococcus lactis* and *Lp. plantarum*. On the other hand, Ivanova et al. (2000) [79] studied the adjustment of the pH of the cell-free supernatant to pH 6.5 and the inhibition of hydrogen peroxide by a catalase treatment. In both cases, the activity was not affected.

Trypsin treatment resulted in the loss of the antimicrobial effect of strains I6, D16, K1 and S4 and decreased the inhibitory effect of the remaining LAB strains (Table 2). The addition of catalase did not affect the suppression by CFS of *Lp. plantarum* D16, L1Q4, S1 and S4 on the growth of *Ps. syringae*. Boiling CFS for 20 min caused a significant decrease in LAB antibacterial activity for strains C6 and Q4. Furthermore, the addition of proteinase K in CFSs led to the loss of antifungal activities of six of the tested LAB and to decreased activity in the remaining three strains, I6, Q3 and S4 (Table 2). Finally, the data strongly suggested that C6 and Q4 produced heat-sensitive bacteriocins. These results showed that the inhibitory effect of strains C6, D16, I6, Q4 and S1 is due to the release of short peptides with antimicrobial activity. The production of heat-stable PlnS and PlnW was reported by Todorov et al. (1999) [80], who observed the highest production of plantaricin S at pH 6. However, in 2005, Todorov [81] reported no differences for plantaricin S production at pH 5.5, 6.0 and 6.5. These observations show that the same LAB species may have different pH requirements for bacteriocin production.

In fact, the successful application of bacteriocins or bacteriocinogenic LAB to eliminate or inhibit the growth of pathogens depends on multiple factors. On one hand, bacteriocin production can be compromised by proteolytic degradation caused by enzymes originating from the product or from endogenous microbiota. On the other hand, the regular exposure of pathogenic bacteria to bacteriocins can induce resistance to these compounds [82]. Antimicrobial peptides against pathogenic bacteria can vary between different genera, identical species and even identical cultures under different environmental conditions.

#### 3.3.2. Antifungal Activity of *Lp. plantarum* Strains against *B. cinerea*

Although much of the literature available is on the antibacterial activity of bacteriocins from LAB, there are only a few reports on their antifungal activity against pathogenic moulds [81]. Manifestation of antimicrobial activity against various pathogenic microorganisms is also an important criterion for the selection of potentially probiotic strains. In the current work, *Lp. plantarum* strains were tested for activity against phytopathogenic *B. cinerea*. The antimicrobial activity of strains C6, D16, I6, Q3, Q4, S1 and S4 against *B. cinerea* was evaluated as high (Table 3). The growth of *B. cinerea* was significantly inhibited by CFSs produced by LAB incubated for up to 48 h. The strongest inhibition effect was detected when CFSs of D16, Q3 and Q4 strains were applied.

After the adjustment of the pH of the supernatants to 6.5 and the addition of catalase, a slight inhibition of the mycelial growth was observed. This indicates that the inhibitory activity of L1 and Q3 strains is due to hydrogen peroxide or acid production. Lavermicocca et al. (2000) [83] reported the isolation of the antifungal compounds phenyl lactic acid and 4-hydroxyphenyl acetic acid from *Lp. plantarum*.

The results with the proteinase K treatment revealed that *Lp. plantarum* L1, Q4 and S1 are producers of proteinaceous substances that greatly suppress the growth of the tested plant pathogen (Table 3). This suggestion is supported by the observed partially decreased antifungal effect after the CFS treatment with proteinase K. Moreover, the antifungal activity of the CFSs from all tested LAB strains was negatively affected by the trypsin treatment. Other authors also reported that the expansion of toxigenic storage fungi was restricted by LAB in vitro and attributed this phenomenon to the combined effect of carboxylic acid and the production of reuterin—a bacteriocin with broad-spectrum antimicrobial activity [83]. Ogunbanwo et al. (2003) [84] examined with an automated turbidimeter and impedimetric methods the in vitro antifungal potential of two *Lp. plantarum* strains, E76 and E98, against different plant pathogenic, toxigenic and gushing active *Fusarium* fungi. *Lp. plantarum* F1 and *L. brevis* OG, isolated from Nigerian fermented food products, produced bacteriocins that had a broad spectrum of inhibition against pathogenic, food-spoilage organisms and various lactic acid bacteria. Zamani-Zadeh et al. (2014) [85] reported similar results with red grapes, in which the combined use of essential oils with fermented MRS by an *Lp. plantarum* strain showed a synergic antifungal effect against *B. cinerea*.

Finally, CFS boiling did not affect the antifungal properties of strains C6, D16, I6, Q3, Q4 and S1. Based on the experimental results, the inhibitory effect against *B. cinerea* may be attributed to the synthesis of thermostable metabolites of a protein nature, capable of suppressing the growth and development of this fungal pathogen.

### 3.4. In Vivo Antimicrobial Activity of Lp. plantarum Q4 against Ps. syringae in Table Grapes

Based on the results from the previous antibacterial activity experiments, strain *L. plantarum* Q4 was selected to study in vivo its activity against *P. syringae* pv. *syringae* in table grapes. Due to the fact that *P. syringae* pv. *syringae* is relatively a newly introduced pathogen in Bulgarian vineyards, symptoms on grapevines may be misidentified as other pathological or physiological conditions. Some symptoms of bacterial inflorescence rot such as necrosis may have previously been attributed to *B. cinerea* [3]. The differences are that *P. syringae* pv. *syringae* infects with a water-soaked appearance and visible bacterial ooze emerging from the plant tissues.

The in vitro studies showed that the CFS from *Lp. plantarum* Q4 had the highest activity against *Ps. syringae* with an inhibition zone of 23.67 ± 0.98 mm (Table 2). In vivo, the antibacterial efficacy of *Lp. plantarum* Q4 against *Ps. syringae* in table grapes showed inhibition on the third, fifth and seventh day after infection (DAI) (Figure 5B). On the seventh DAI with *Ps. syringae*, necrosis was observed with visible bacterial ooze from the plant tissues (Figure 5A) and bacterial infection on the wounded fruits progressed along with the tissues (Figure 5a). In the absence of the *Lp. plantarum* Q4 CFS, the bacterial pathogen caused 100% infection with lesions of the fruits (Figure 5a). In the presence of the *Lp. plantarum* Q4 CFS, a decrease in fruit infection up to 45% was observed on the seventh DAI. Wounded control grapes (Figure 5e) and control grapes (Figure 5d) had no visible symptoms of the disease, as well as grapes after treatment with the *Lp. plantarum* Q4 CFS (Figure 5c). The results demonstrate a promising potential for the suppression of the *Ps. syringae* infection of grapes and expand the knowledge on antimicrobial applications of CFSs from *Lp. plantarum* strains. These findings indicate that LAB are good candidates for the development of microbial biopesticides since they are categorised as GRAS. The data obtained from experiments with table grapes, both in vitro and in vivo, show that the treatment with CFS of *Lp. plantarum* Q4, which is a producer of several different plantaricins, can be used to control *P. syringae* subsp. *syringae*.

### 3.5. In Vivo Antimicrobial Activity of Lp. plantarum Q4 against Botrytis cinerea in Grapes

Grey rot on grapes caused by *B. cinerea* is an economically relevant fungal disease of table grapes. Under the conditions of high humidity and temperature, grey rot affects buds, young shoots and leaves, but more severely, the grapes. In some years, grey rot may destroy up to 60–70% of the harvest of susceptible grape varieties. The skin of diseased grapes acquires a light-brown colour, peels easily and cracks. At first, small brown spots appear, which gradually increase and cover the whole grape. Under favourable conditions, the disease is transmitted to neighbouring grapes, covering the entire bunch with mycelia. It is of key importance to find ecologically friendly means to control this pathogen, which is one of the objectives of the present study.

Figure 6 shows the antifungal effect of the CFS from *Lp. plantarum* Q4 against *B. cinerea* compared to the treatment with the control wounded grapes treated with MRS only (Figure 6d,e), wounded grapes treated with the *Lp. plantarum* Q4 fermented CFS (Figure 6c), wounded grapes treated with both the *Lp. plantarum* Q4 CFS and *B. cinerea spores (*Figure 6b), and wounded grapes infected with *B. cinerea* spores (Figure 6a). Results were examined on the seventh DAI. At this time, a decrease in the number of contaminated grapes was achieved in the *Lp. plantarum* Q4 CFS- and *B. cinerea* spore-treated fruits—from 85% to 10% less contaminated grapes compared to *B. cinerea* spore-treated fruits (Figure 6B). Our results are similar to those reported by Dupozo et al. (2022) and Marín et al. (2019) [86,87].

Previous reports suggested that *Lp. plantarum* had the ability to inhibit pathogenic fungi, including genera of *Pectobacterium* and *Penicillium*, on Chinese cabbage and in orange juice [88,89]. *Lactiplantibacillus plantarum* IMAU10014 isolated from koumiss produced a broad spectrum of antifungal compounds, all of which were active against plant pathogenic fungi [90]. *Weissella cibaria* and *Weissella paramesenteroides* could inhibit four fungal species of *Penicillium oxalicum*, *Asperillus flavus*, *Aspergillus sydowii* and *Mucor racemosus* on grapes [91]. Similar to our results, Chen et al. 2020 [92] reported that *Lactiplantibacillus plantarum* CM-3 demonstrated the highest potential for in vitro and in vivo control of *B. cinerea* by the reduction of the mycelial growth and spore germination of *B. cinerea*. These results suggested that applying bacteriocin producing *Lp. plantarum* Q4 as a biological agent to control fungal infection of fruit and vegetables will have an essential significance in the near future.

The present study shows that *Lp. plantarum* Q4 can significantly suppress the growth of the phytopathogenic fungus *B. cinerea* and bacteria *Ps. syringae*. The in vivo experiment demonstrated that fungal contamination in table grapes is reduced after treatment with CFSs from lactic acid bacteria. *Lp. plantarum* Q4 showed great potential as a biocontrol agent for postharvest diseases of table grapes, especially when combined with cold storage. Current experiments indicate that *Lp. plantarum* Q4 produces thermostable and active compounds with an acidic or proteinaceous nature. Therefore, this strain would be an effective agent for grape preservation during storage.

## 4. Conclusions

In recent decades, consumers’ increasing interest in the use of natural food preservatives has been observed. Table grapes are one of the most economically important fruits in Bulgaria, but, unfortunately, they are highly susceptible to different fungal and bacterial pathogens. Research is progressing to find new ecologically friendly means to control such diseases, and lately, much attention has been focused on microbial bacteriocin producers and their potential applications. Nine bacteriocin-producing *Lp. plantarum* strains isolated from sourdoughs were screened for their ability to inhibit the growth of *Ps. syringae* pv. *syringae* and *B. cinerea*, which cause significant economic losses in table grape production and the trade chain. The PCR amplification and DNA sequencing proved that the tested bacteriocin-producing LAB belonged to *Lactiplantibacillus plantarum*, which is very common in plant-based fermented foods. A significant inhibitory effect of the studied *Lp. plantarum* strains against the tested plant pathogens was observed. The presence of several *pln* genes in the genome of the tested *Lp. plantarum* strains, including *pln*S and *pln*W encoding thermostable plantaricins, showed that these strains are new multibacteriocin producers with the potential for applications in the development of eco-friendly products for plant protection and biopreservation of table grapes.

## Figures and Tables

**Figure 1 microorganisms-10-02094-f001:**
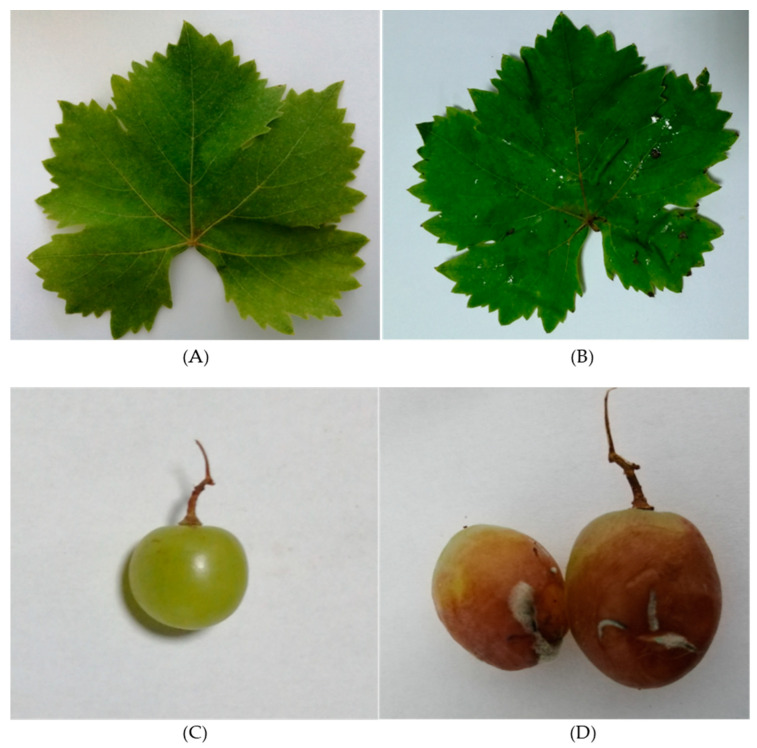
Confirmation of *Ps. syringae* pv. *syringae* and *B. cinerea* infection on leaves and fruits of *V. vinifera.* (**A**) Control grape leaves; (**B**) *Ps. Syringae* pv. *Syringae*-infected leaves; (**C**) control grapes; (**D**) *B. cinerea* infection on grapes.

**Figure 2 microorganisms-10-02094-f002:**
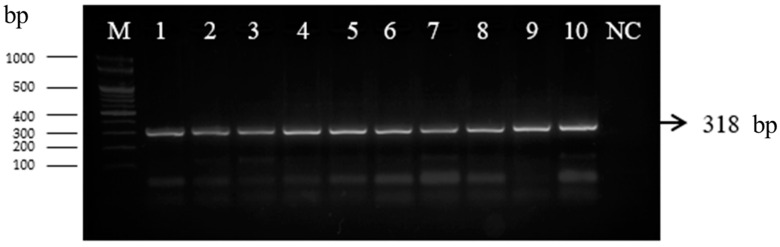
PCR amplification of nine *Lb. plantarum* strains from Bulgarian sourdoughs with *recA* gene primers (lanes 1 to 9) as follows: Lane 1—*Lp. plantarum* C6; Lane 2—*Lp. plantarum* D16; Lane 3—*Lp. plantarum* I6; Lane 4—*Lp. plantarum* K1; Lane 5—*Lp. plantarum* L1; Lane 6—*Lp. plantarum* Q3; Lane 7—*Lp. plantarum* Q4; Lane 8—*Lp. plantarum* S1; Lane 9—*Lp. plantarum* S4; Lane 10—reference strain LP-*Lb. plantarum* ATCC 14917 T; NC—negative control without DNA.

**Figure 3 microorganisms-10-02094-f003:**
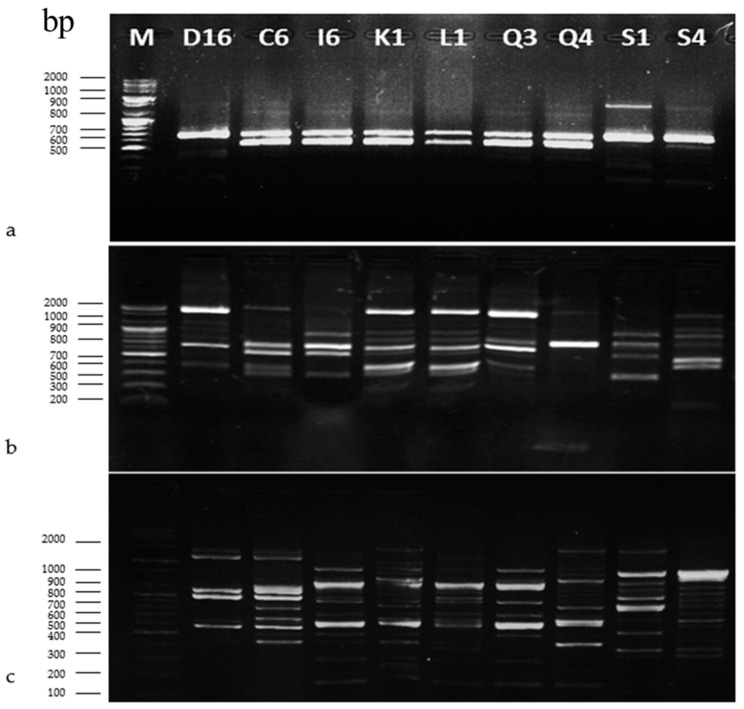
Electropherograms of DNA fragments obtained by RADP-PCR analysis of isolated *Lp. plantarum* strains with primers (**a**) P1, (**b**) P4 and (**c**) M13. M—molecular marker; Lane 2÷10—*Lp. plantarum* strains.

**Figure 4 microorganisms-10-02094-f004:**
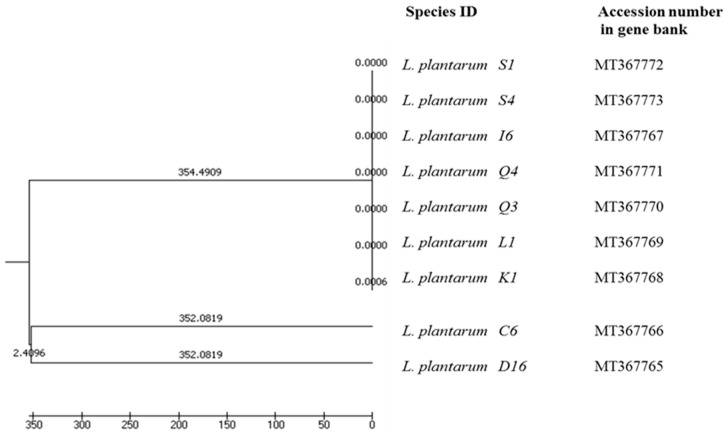
An UPGMA dendrogram of phylogenetic similarity of the studied *Lp. plantarum* strains.

**Figure 5 microorganisms-10-02094-f005:**
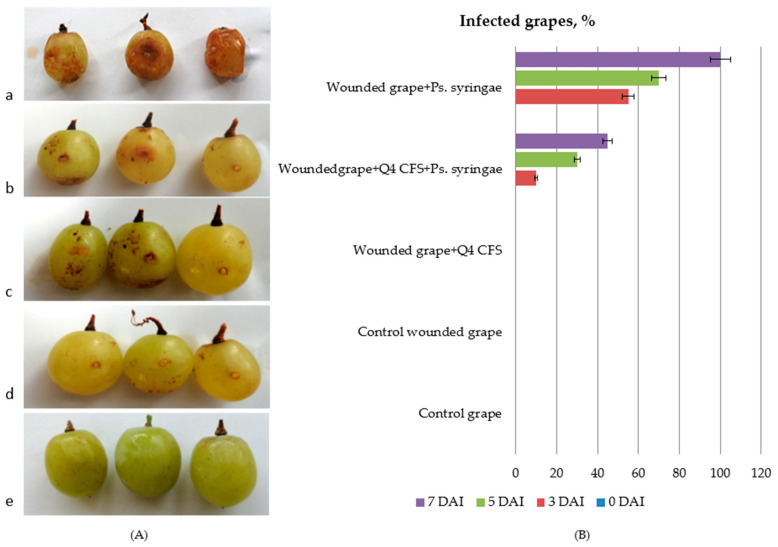
In vivo antimicrobial activity of *Lp. plantarum* Q4 against *Ps. syringae* in grapes. (**A**) In vivo infection of grapes with *Ps. syringae* on the seventh DAI: (**a**) 100% wounded grapes + *Ps. syringae*; (**b**) 45% wounded grapes + CFS from *Lp. plantarum* Q4 + *Ps. syringae*; (**c**) wounded grapes + CFS from *Lp. plantarum* Q4; (**d**) control wounded grapes; (**e**) control grapes. (**B**). In vivo antifungal effect of CFS from *Lp. plantarum* Q4, expressed as percent of infected grapes.

**Figure 6 microorganisms-10-02094-f006:**
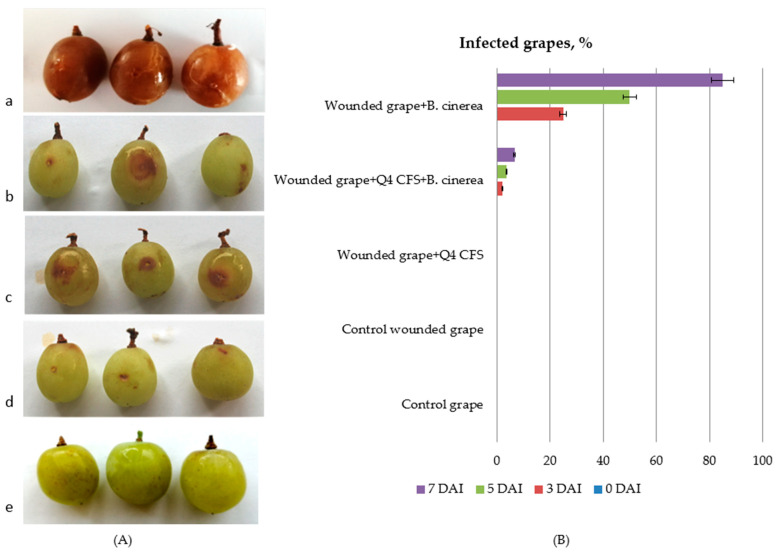
In vivo antimicrobial activity of *Lp. plantarum* Q4 CFS against *B. cinerea* in table grapes. (**A**) In vivo infection of grapes with *B. cinerea*: (**a**) 85% infection of wounded grapes + *B. cinerea*; (**b**) 6.6% wounded grapes + *Lp. plantarum* Q4 CFS + *B. cinerea*; (**c**) wounded grapes + *Lp. plantarum* Q4 CFS; (**d**) control wounded grapes; (**e**) control grapes. (**B**). In vivo antifungal effect of *Lp. plantarum* Q4 CFS, expressed as percentage of infected grapes.

**Table 1 microorganisms-10-02094-t001:** Amplified plantaricin (*pln*) genes in the genome of *Lp. plantarum* strains isolated from spontaneously fermented sourdoughs.

Bacteriocins/ *Lp. plantarum* Strains	C6	D16	I6	K1	L1	Q3	Q4	S1	S4
*plnA*	+	ND	ND	+	+	+	+	+	+
*plnB*	+	+	+	+	+	+	+	+	+
*plnC*	+	ND	+	+	ND	+	+	+	+
*plnD*	+	+	+	+	+	+	+	+	+
*plnEF*	+	+	+	+	+	+	+	+	+
*plnG*	+	+	+	+	+	+	+	+	+
*plnI*	+	+	+	+	+	+	+	+	+
*plnJ*	+	+	ND	+	+	+	+	+	+
*plnK*	+	ND	ND	ND	+	+	ND	+	+
*plnN*	+	+	+	+	+	+	+	+	+
*plnNC8*	+	+	+	+	+	+	+	ND	+
*plnS*	ND	+	+	+	+	+	+	ND	+
*plnW*	ND	+	ND	ND	ND	ND	ND	ND	ND

Note: ND—not detected, +—positive result.

**Table 2 microorganisms-10-02094-t002:** Antibacterial activity of *Lp. plantarum* strains against *Ps. syringae* pv. *syringae* by agar diffusion assay.

LAB Strain	Treatment					
	CFS	Trypsin	pH	Proteinase K	Catalase	Boiling
	Inhibition Zone, mm					
**C6**	16.00 ± 1.02 d**	3.10 ± 0.79 c	1.17 ± 0.36 b	0.00 a	0.00 a	3.03 ± 0.33 c**
**D16**	23.67 ± 0.93 d**	0.00 a	0.00 a	0.00 a	18.00 ± 1.36 d**	0.00 a
**I6**	16.00 ± 0.85 d**	0.00 a	2.03 ± 0.26 d**	1.67 ± 0.59 b	0.00 a	0.00 a
**L1**	12.17 ± 0.76 c	26.67 ± 0.92 e**	0.00 a	0.00 a	1.33 ± 0.23 b	0.00 a
**K1**	2.33 ± 0.65 a	0.00 a	1.63 ± 0.37 c	0.00 a	0.00 a	0.00 a
**Q3**	15.67 ± 0.83 d**	5.33 ± 0.46 d**	1.77 ± 0.59 cd**	2.50 ± 0.74 c**	0.00 a	0.00 a
**Q4**	23.67 ± 0.98 e**	3.17 ± 0.85 c	2.53 ± 0.83 e**	0.00 a	2.33 ± 0.47 c**	1.90 ± 0.42 b**
**S1**	5.80 ± 0.57 b	2.00 ± 0.45 b	2.00 ± 0.72 d**	0.00 a	2.23 ± 0.49 c**	0.00 a
**S4**	1.70 ± 0.32 a	0.00 a	0.00 a	1.330 ± 0.46 b	1.70 ± 0.39 bc	0.00 a

**Note**: **CFS**—cell-free supernatant, **Trypsin**—CFS + trypsin of 1 mg/mL, **pH**—pH adjusted from 3.5–4.2 to 6.5, **Proteinase K**—CFS + proteinase K (100 U/mg), **Catalase**—CFS + catalase (5 mg), **Boiling**—boiling of CFS for 20 min. The antibacterial activity was expressed as follows: strong (10 mm of inhibition zone between the well and fungal growth); a = week growth (less than < 3 mm); b = medium inhibition (>8 mm); c = strong inhibition (>10 mm of inhibition zone). Means with different letters (a–e) in the same column differ at a *p* < 0.05 level of probability by the LSD test; ± SD—standard deviation; ** indicates high significance (*p* < 0.05).

**Table 3 microorganisms-10-02094-t003:** Antifungal activity of *Lp. plantarum* strains against *B. cinerea*.

LAB Strain	Treatment					
	CFS	Trypsin	pH	Proteinase K	Catalase	Boiling
	Mycelial Growth, %					
**C6**	81.62 ± 1.32 d	45.86 ± 2.15 a**	72.87 ± 1.26 c	82.04 ± 0.99 d**	68.53 ± 1.28 c**	49.47 ± 1.02 b**
**D16**	83.34 ± 1.50 d	81.94 ± 0.61 d	74.37 ± 1.76 c	83.04 ± 0.58 d	80.93 ± 0.93 cd	46.45 ± 0.97 ab
**I6**	82.04 ± 1.93 d	64.65 ± 1.00 d	72.51 ± 0.87 c	71.33 ± 1.27 c	79.28 ± 0.63 cd	62.29 ± 1.38 ab
**L1**	82.22 ± 0.57 d	82.51 ± 1.26 d	68.51 ± 2.01 cd**	52.43 ± 0.72 b	45.86 ± 0.98 b**	70.58 ± 0.88 c
**K1**	79.61 ± 1.30 cd**	55.24 ± 2.04 b	64.11 ± 1.52 c	71.48 ± 1.55 c	78.12 ± 1.30 cd	72.09 ± 1.12 c
**Q3**	82.87 ± 0.81 d	78.37 ± 1.99 cd**	58.97 ± 0.73 b**	60.35 ± 0.73 bc**	65.14 ± 1.22 c**	30.31 ± 0.75 a**
**Q4**	83.42 ± 1.28 d	52.24 ± 1.63 b**	67.57 ± 0.65 c	31.93 ± 1.28 a	73.06 ± 0.96 c**	30.15 ± 1.54 a**
**S1**	80.93 ± 0.668 d	66.35 ± 2.04 cd**	79.62 ± 1.00 cd**	23.75 ± 1.49 a	52.35 ± 1.66 b**	49.45 ± 0.73 b**
**S4**	81.21 ± 1.53 d	82.92 ± 0.94 d	71.82 ± 0.92 c	69.28 ± 0.94 bc**	68.45 ± 1.11 bc**	73.36 ± 1.62 c

**Note**: **CFS**—cell-free supernatant, **Trypsin**—CFS + trypsin of 1 mg/mL, **pH**—pH adjusted from 3.5–4.2 to 6.5, **Proteinase K**—CFS + proteinase K (100 U/mg), **Catalase**—CFS + catalase (5 mg), **Boiling**—boiling of CFS for 20 min. The antibacterial activity was expressed as percentage of radial mycelial growth of the phytopathogenic fungus. SD—standard deviation; a–d indicate statistical differences; ** indicates high significance (*p* < 0.05).

## Data Availability

The data are available from the corresponding author, upon reasonable request.

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
