# Peer review of "Screening of Lactiplantibacillus plantarum Strains from Sourdoughs for Biosuppression of Pseudomonas syringae pv. syringae and Botrytis cinerea in Table Grapes"

_microorganisms, 2022, doi:10.3390/microorganisms10112094_

Round 1

Reviewer 1 Report

Somewhere, the format should be modified, such as the underlines in line 130-131 should be removed, In vivo should be italic, the size of words is inconsistent.

In section 2.3.3, what are the primers used for the PCR of plantaricins from plnA to plnW?

Authors emphasized bacteriocins in this study, but cell free supernatant was used in in vivo assay. As we known, bacteriocin in the cell free supernatant contributed only part of the antimicrobial activity. The cell free supernatant has organic acid, hydrogen peroxide, diacetyl, etc., they also had antimicrobial activity.

In Fig. 2 and Fig. 3, label the size of DNA Marker

The reference lists must be revised with too many errors.

Author Response

Dear Reviewer, thank you for the comments and remarks regarding our manuscript. We did our best to respond and improve the article. Please find below our answers:

Point 1. There is obviously a problem with the MDPI software which converts the doc file into a pdf file during the process of manuscript uploading. We do not see the pdf at the point of uploading the manuscript. So, during this conversion, the original formatting of the doc file is messed up, which is the reason for most editing remarks for errors (missing italic, appearing underlined words, highlighted, references not according the journal’s requirements) which are not present in the original doc file which we submit. So, we were also very surprised to see these formatting changes in the pdf version. Please, during the second revision, download also the doc version for comparison.

Point 2. The primers which we used were according the article of Omar et al. (2008), that is why in section 2.3.3 we have provided a reference to this article.

Point 3. The antimicrobial activity of LAB is, indeed, due to multiple factors, among which organic acids, hydrogen peroxide and bacteriocins of proteinaceous nature. In order to understand which of these factors has a major effect depending on the LAB strain and the tested pathogen, we experimented with different treatments of the cell-free supernatant. In one trial line we used pH adjustment to 6.5 to eliminate the effect of organic acids, in another trial line we used catalase to eliminate the effect of hydrogen peroxide; trypsin, boiling and the use of proteinase K were used as various treatments aiming to eliminate the effect of proteinaceous compounds such as bacteriocins. So the different treatments aimed to clarify which antimicrobial agent had the most significant effect from the various LAB strains and on the different pathogens.

Point 4. The DNA markers in Figs. 2 and 3 are labelled, please see the uploaded file with revised figures.

Point 5. The errors in the reference list only appear in the pdf file and are obviously caused by the converting software in the journal’s portal. For comparison, please see the doc file.     

Reviewer 2 Report

Line 46: antimicrobial
Lines 159-160: g/L
Line 223: Lb. plantarum in italics
Line 262: B. cinerea in italics
Line 271: "agar dilution" why is it underlined?
Line 286: "Tryfinopoulou et al. (2015)" why is it underlined?
Line 291: "In vivo" and "Lp. plantarum" in italics
Line 392: "In vivo", "Lp. plantarum" and "B. cinerea" in italics
Lines 328-336: Comment on fig. 1a, or remove it from fig. 1.
Lines 497: "Lp. plantarum" and "B.cinerea" in italics
Fig. 5a: the berries are crushed you need to revise the proportions of the photo
Line 732: eliminate red highlighting
Line 753: eliminate red highlighting
References: Perform a general check of the citations included in the article. In some cases they were not reported following the rules for authors. Check especially the italicized name of microbial species, and also the journal abbreviation.

Author Response

Dear Reviewer, thank you for the comments and remarks regarding our manuscript. We did our best to respond and improve the article. Please find below our answers:

 Line 46: revised

Lines 159-160: revised

Lines 328-336: revised

Fig 5 and Fig 6: the photos are revised, please see the uploaded file with revised figures.

Remarks regarding editing: There is obviously a problem with the MDPI software which converts the doc file into a pdf file during the process of manuscript uploading. We do not see the pdf at the point of uploading the manuscript. So, during this conversion, the original formatting of the doc file is messed up, which is the reason for most editing remarks for errors (missing italic, appearing underlined words, highlighted, references not according the journal’s requirements) which are not present in the original doc file which we submit. So, we were also very surprised to see these formatting changes in the pdf version. Please, during the second revision, download also the doc version for comparison.

We re-typed many of the problematic paragraphs just in case, but we think that the same problem will occur again when we upload the revised manuscript. Again, please also see the doc format of the manuscript.

Round 2

Reviewer 1 Report

No